# Respiratory Muscle Strength in Brazilian Adolescents: Impact of Body Composition

Viviane Campos de Lima [1], Marcelo Luis Marquezi [1,2], Paulo Roberto Alcantara [1], Nayara Barbosa Lopes [1,3], Caroline Santana Frientes [1,2], Thais Miriã da Silva Santos [1,3], Leonardo Ribeiro Miedes [1,3], Matheus Silva Fornel [1], Danielle Castro Oliveira [1], Patrícia Soares Rogeri [2], Antônio Herbert Lancha Junior [2], Nathalia Bernardes [1,3] and Juliana Monique Lino Aparecido [1,2,*]

[1] Laboratory of Physical Education Research (LAPEF), University City of Sao Paulo (UNICID), Sao Paulo 05508-030, Brazil; nbernardes@outlook.com (N.B.)
[2] Laboratory of Clinical Investigation: Experimental Surgery (LIM 26), Clinic's Hospital of Medical School, University of Sao Paulo, Sao Paulo 05508-030, Brazil; lanchajunior@gmail.com (A.H.L.J.)
[3] Human Movement Laboratory (LMH), University San Judas Tadeu (USJT), Sao Paulo 03166-000, Brazil
[*] Correspondence: jmonique.lino@gmail.com; Tel.: +55-(11)9-6997-5162

**Abstract:** (1) Introduction: Studies on respiratory muscle strength (RMS) in adolescents are controversial. Few studies so far have investigated respiratory muscle strength in Brazilian adolescents and the impact of body composition on it. (2) Objective: to evaluate the respiratory muscle strength of Brazilian adolescents and compare this with nationally and internationally predicted normality values. (3) Method: A cross-sectional study (CAEE: 34634414.5.0000.5479) was carried out with 98 adolescents, where both sexes were divided into four groups: eutrophic (n = 44); overweight (n = 15), obese (n = 25), and severely obese (n = 14). All were submitted to an anthropometric assessment, body composition analysis and manovacuometry. To interpret the results, Analysis of Variance (ANOVA) with Turkey's post hoc test was used. The Kruskal–Wallis test and Friedman's post hoc test were used to compare the observed vs. proposed results. A *p*-value < 0.05 was considered statistically significant. (4) Results: There were no differences among the groups for maximal inspiratory (MIP) and expiratory (MEP) pressures. However, when considering the total sample, we could say that RMS was higher among boys, and there were no significant differences in RMS in relation to the maturational stage. The values obtained for MIP were lower than those suggested for the national equation and higher than those proposed for the international equation. Similarly, the values obtained for MEP were lower than those suggested for the national and international equation. (5) Conclusions: RMS was similar in adolescents with different body compositions and different maturation stages. Adiposity did not interfere with RMS in adolescents. Boys had higher MIP and MEP values compared to girls. Therefore, the reference values proposed by the equations do not consistently match RMS in the adolescents studied. This context reinforces the need for new studies that are related to RMS to establish normality values and propose equations that represent the youth population.

**Keywords:** obesity; adolescent; reference values; respiratory muscle strength

## 1. Introduction

Childhood obesity is not merely a change in body composition but a disorder of metabolism that occurs through the interaction of genetic, environmental, and behavioral factors leading to an excessive accumulation of adipose tissue, which is usually related not only to a higher caloric intake but also a lower energy expenditure [1].

Appointed as the most prevalent metabolic disease in the world, obesity reaches epidemiological proportions and is directly related to disorders of low self-esteem, impaired body image, depression, and decreased quality of life [2]. In Brazil, obesity among children and adolescents aged 5 to 19 years increased between 1975 and 2016, from 0.9% to 12.7% in boys

and from 1% to 9.4% in girls. According to Pazzianotto Filho et al. [3], this growth increases the risk factors for cardiovascular diseases, type 2 diabetes, rheumatoid arthritis, neoplasms, and respiratory diseases (sleep apnea and hypoventilation syndrome) in adult life [4].

Santiago et al. [5] emphasized that obesity leads to changes in respiratory mechanics that are caused by the accumulation of fat in the ribs, diaphragm, and abdomen, reducing rib cage compliance and causing less diaphragmatic excursion in this population of overweight/obese children and adolescents.

According to Tenório et al. [6], the consequent increase in oxygen consumption leads to several factors, such as exercise intolerance, changes in gas exchange, the control of breathing patterns, and the strength and resistance of respiratory muscles, which is commonly observed in obese adults.

There are several invasive or non-invasive methods to assess respiratory muscle strength, with manovacuometry being the non-invasive method most frequently used in clinical practice. Despite its low cost, manovacuometry can verify maximal inspiratory and expiratory muscle forces. It measures intraoral pressure during a maximal inspiratory (MIP) or an expiratory (MEP) effort against the occluded airway. MIP assesses the strength of the inspiratory muscles, while MEP evaluates the strength of the expiratory muscles [7]. Bessa et al. [8] added that these measurements are important in the assessment of respiratory muscle strength in different conditions, such as in healthy individuals of different age groups, in patients with disorders of different origins, in the pre and postoperative period of thoracoabdominal surgeries, and respiratory muscle training.

Given the importance of measuring respiratory muscle strength, many studies have emerged to establish predictive equations and tables of reference values, considering factors such as age, sex, and height in different populations. However, Laranjeira et al. [9] added that the equations for such predicted values, when applied in daily practice, do not always correlate with the values obtained, which according to Teixeira et al. [10], may be associated with the inverse relationship between adiposity and respiratory muscle strength. In the face of what has been found, and aware that the association between adiposity and respiratory muscle strength is still a controversial topic that is not well elucidated in children and adolescents, the present study aimed to evaluate the respiratory muscle strength of Brazilian adolescents and compare this with nationally [11] and internationally [12] predicted normality values.

## 2. Materials and Methods

### 2.1. Trial Design, Setting and Ethics

This is a cross-sectional study where adolescents of both sexes were allocated into four groups: eutrophic (EG), overweight (OwG), obese (OG), and severely obese (SOG). After parents/guardians and adolescents, respectively, signed the Terms of Free and Informed Consent and Assent (CAEE: 34634414.5.0000.5479), these adolescents were submitted to a pubertal evaluation, physical activity level, anthropometry, body composition, and respiratory muscle strength. We reported the study methods and results in accordance with the Strengthening the Reporting of Observational Studies in Epidemiology (STROBE) reporting guideline [13].

### 2.2. Sample Size

For the sample calculation, MIP and MEP data were analyzed for 73 adolescents obtained in the pilot study by Lopes et al. [14]. In order to make such a calculation, these data were compared with the values presented in the literature, considering a difference of 30.00 cm $H_2O$ to be detected [15,16]. An analysis of covariance (ANCOVA) was used to adjust the covariates: age, sex and practice of physical activity, with a standard deviation of 27.65 cm $H_2O$ and 22.67 cm $H_2O$, for MIP and MEP, respectively. A test power of 95% with a significance level of 5% was used; 16 and 11 individuals were estimated as sufficient to compose the sample in each group, i.e., MIP and MEP, respectively.

*2.3. Participants*

We recruited adolescents for this study in a hospital and four municipal schools located in the metropolitan region of a Brazilian capital city from May 2017 to January 2019. We worked with school and hospital boards (mainly deans/principals) to obtain their consent to collaborate, and then a trained multidisciplinary team (doctor, nurse, physical therapist, and physical education teachers) visited the pediatric clinic and schools to give presentations on the benefits of good health. After these educational sessions, we evaluated all adolescents from these institutions for weight and height.

We included in this study, adolescents aged between 12 and 16 years, and both sexes were allocated into four groups, as already mentioned before eutrophic (EG), overweight (OwG), obese (OG), and severely obese (SOG), according to the Z-score curves of the Body Mass Index (zBMI) proposed by the World Health Organization (WHO), including the body composition [17,18]. These boys and girls had to present minimum physical fitness to be included in this study. Therefore, we initially classified them according to the International Physical Activity Questionnaire (IPAQ) and according to the activity performed during the last week. The participant was considered active when he/she performed five days or more of physical activity with a total of 150 min or more per week; irregularly active when not reaching the 150 min threshold; or sedentary when performing 10 min or less per week [19]. As all children included participated in physical activities at school, none were considered sedentary in this study, and all were fit.

We excluded individuals who lived in cities or states outside the coverage of the free transportation offered or with a time conflict between the sessions and the work of their companions or those undergoing drug treatment for weight control or who had heart, orthopedic, respiratory, or cardiac problems, including kidney disease, diabetes, uncontrolled hypertension, genetic syndrome, or hormonal abnormalities.

Thus, 445 adolescents were recruited, but 347 were excluded for the following reasons: 202 did not meet the inclusion criteria, and 141 refused to participate due to logistical issues, such as living elsewhere or having no companions; therefore, 98 adolescents participated in the present study (Figure 1).

The maturational evaluation was obtained from the self-assessment of maturation, based on Tanner's plank (Portuguese version), in a separate room, with the previous guidance of the participants and with the support of illustrative images of breasts, genitals, and pubic hair in each maturational stage. The adolescents identified the stage that most closely approximated their personal image. The girls were evaluated by their breasts and hairless characteristics; the boys were assessed through the development of gonads and hairiness in the genital region. They were classified as I prepubescent state, II, III, IV pubescent, and V post-pubertal [20].

Their body composition was determined by bioimpedance (Biodynamics 310, TBW Importer Ltda, Sao Paulo, Brazil). Before the evaluation, the participants underwent the following conditions: 24 h of hydration with good quality water; the prohibition of the consumption of any alcoholic drinks or stimulants; 12 h without the practice of strenuous physical activity; their bladders would have to be empty before the evaluation; the girls could not be menstruating [7–12,16–21].

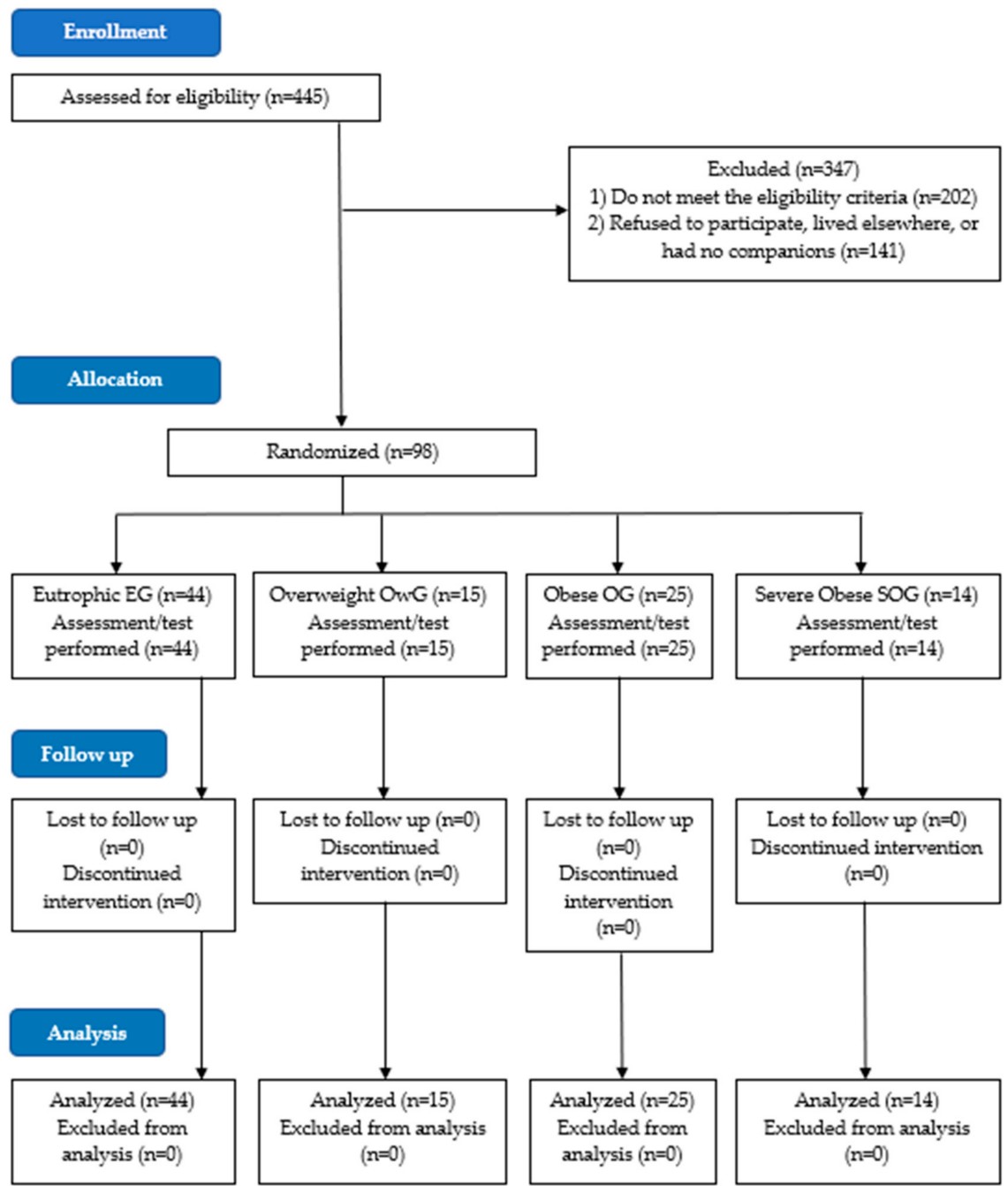

**Figure 1.** Flow chart STROBE of recruitment and randomization of the analyzed groups.

*2.4. Anthropometric Assessment*

Body mass and height were measured on a mechanical scale with a stadiometer (Filizola®, Toledo do Brasil Indústria de Balanças Ltda, Sao Paulo, Brazil), a capacity of up to 150 kg, an accuracy of 100 g and a scale in meters. For this evaluation, the subjects were instructed to be barefoot, with light clothes, with their head and arms aligned to the body. The values obtained were used to calculate BMI (Body Mass Index), given by the weight (in kilograms)/height ratio (in meters squared). The zBMI was calculated using the following formula: Z-score = (BMI observed) − (reference median BMI)/Population Standard Deviation, using the WHO AnthroPlus® Software package (version 1.0.4, WHO, Geneva, Switzerland), proposed by WHO [17].

*2.5. Assessment of the Level of Physical Activity*

The level of physical activity was evaluated using the International Physical Activity Questionnaire (IPAQ) in its short version [22]. The place and time spent while the participants were performing their physical activity during a week were considered, including the practice of activities in different contexts of daily life, such as housework, leisure and transportation, as well as passive activities, i.e., those performed while the individuals were sitting. This level can be classified into four categories: very active, active, irregularly active and sedentary. Therefore, the level of physical activity was considered as follows [22]:

- Very active: all those who practiced in vigorous activity $\geq 3$ days a week and $\geq 20$ min per session plus moderate activity and/or walking $\geq 5$ days a week and $\geq 30$ min per session;
- Active: those who performed an activity $\geq 5$ days a week and $\geq 150$ min a week (walking plus moderate activity plus vigorous activity).
- Irregularly active: those who performed physical activity but not enough to be classified as active, as did not comply with the frequency or duration of exercise in different types of activities (walking plus moderate activity plus vigorous activity).
- Sedentary: those who did not perform any physical activity for at least 10 min continuously during a week.

*2.6. Maximum Respiratory Pressures*

By employing an analog manovacuometer (Commercial Medical Ltd., Sao Paulo, Brazil) with a variation of 4 in 4 cm $H_2O$ and a pressure analysis limit of $0 \pm 150$ cm $H_2O$, the values of MIP and MEP were measured. Following the orientation of the Consensus of Respiratory Muscle Tests, measurements were performed with the participants seated, with the chest and feet supported, and with the use of a nasal clip. In this position, adolescents were instructed to place the nasal clip and hold the manovacuometer by tightening the mouth firmly against the lips, preventing air leakage, to perform maximal inspiration (from the residual volume) for the MIP measurement, and maximum expiration (from the total lung capacity) to determine the MEP. Three consecutive measurements of MIP and MEP were performed with a rest of 30 to 60 s between them, and the highest value obtained was used [7,23].

*2.7. Statistical Analysis*

For statistical analysis, the software JAMOVI (version 1.6.23.0, Sydney, Australia) was used. The parametric data were presented in relative and absolute frequency with the mean and standard deviation of the mean; non-parametric data were described as the median and percentiles of 25 and 75. Normality was verified using the Shapiro–Wilks test.

For comparisons of the obtained values among the groups, variance analysis tests (ANOVA) with Tukey's post hoc tests were used for parametric data, and the Kruskal–Wallis test (observed vs. proposed) with the post hoc analysis of Friedman's test for nonparametric data. For the analysis of significance in percentage values among the groups, the percentage variation ($\Delta\%$) was calculated using the equation (($V2 - V1$)/$V1 \times 100$), in which V1 represents the initial value among the groups, and V2 is the final low. The significance level adopted was $p < 0.05$ for all analyses.

## 3. Results

Ninety-eight adolescents participated in the study, including 60 females (59.10%) and 38 males (38.9%), with a mean age of $13.02 \pm 0.95$ years and were allocated into four groups, according to the statistical differences observed in the body composition data (Table 1). The maturational state with the highest prevalence was prepubertal (EG: 45.5%; OwG: 93.3; OG: 76%; SOG: 42.8%), and the level of physical activity with the highest prevalence was active (EG: 54.5%; OwG: 0.0%; OG: 52%; SOG: 43.0%).

**Table 1.** Adolescents' characterization: body composition. São Paulo-SP, 2019.

| Variable | EG (n = 44) | OwG (n = 15) | OG (n = 25) | SOG (n = 14) |
|---|---|---|---|---|
| Age (years) [b] | 13.0 (12.0–14.0) | 13.0 (13.0–14.0) | 13.0 (12.0–13.0) | 12.5 (12.0–15.0) |
| Weight (kg) [a] | 46.2 ± 8.67 | 53.5 ± 8.95 | 68.2 ± 9.90 *≠ | 74.5 ± 13.9 *≠ |
| Height (m) [a] | 1.6 ± 0.0936 | 1.6 ± 0.0523 | 1.6 ± 0.0551 | 1.6 ± 0.0838 |
| pBF (%) [a] | 20.6 ± 5.47 | 26.0 ± 2.56 * | 31.3 ± 2.69 *≠ | 38.0 ± 2.73 *≠¥ |
| FM [b] | 8.8 (6.75–12.1) | 13.9 (12.5–15.7) * | 19.5 (17.8–23.9) *≠ | 29.8 (25.6–39.9) *≠¥ |
| LM [a] | 36.1 ± 8.46 | 39.7 ± 7.20 | 44.5 ± 7.66 * | 46.7 ± 8.77 * |
| BMR [b] | 1084 (967–1193) | 1151 (1076–1311) | 1370 (1221–1511) *≠ | 1460 (1281–1596) * |
| BMI (kg/m²) [b] | 18.3 (17.0–21.0) | 19.8 (19.2–22.7) * | 26.9 (24.4–28.2) *≠ | 29.2 (26.6–31.6) *≠ |

[a] Indicates mean values and standard deviation of the mean; [b] Indicates median values and percentiles 25 e 75; EG:eutrophic group; OwG: overweight group; OG: obese group; SOG: group of severely obese; Absolute frequency values (n); weight (kg); height (m); pBF: body fat percentage (%); FM: fat mass; LM: Lean mass; BMR: basal metabolic rate (cal); BMI: Body mass index (kg/m²): * indicates $p < 0.05$ vs. EG; ≠ indicates $p < 0.05$ vs. OwG; ¥ indicates $p < 0.05$ vs. OG. n = 98.

When comparing the values obtained for maximum respiratory pressures among the groups, no differences were observed (Table 2).

**Table 2.** Assessment of respiratory muscle strength: MIP and MEP values obtained and proposed by the Lanza et al. [11] and Verma et al. [12] (n = 98), Sao Paulo-SP, 2019.

| Pressures (cm H₂O) | EG (n = 44) [b] | OwG (n = 15) [b] | OG (n = 25) [b] | SOG (n = 14) [b] |
|---|---|---|---|---|
| MIP_obtained | 80.0 (60.0–106.5) | 60.0 (44.0–100.0) | 68.0 (60.0–90.0) | 76.0 (56.0–80.0) |
| MIP_LANZA et al. [11] | 99.6 (99.6–109.5) | 99.6 (99.6–109.5) | 99.6 (99.6–109.5) | 99.6 (99.6–99.6) |
| MIP_VERMA et al. [12] | 70.0 (64.9–74.5) | 73.7 (64.5–78.4) | 78.7 (57.8–84.2) | 84.62 (79.3–93.9) *≠ |
| MEP_obtained | 76.0 (58.0–92.5) | 64.0 (60.0–80.0) | 64.0 (56.0–92.0) | 69.0 (44.0–80.0) |
| MEP_LANZA et al. [11] | 96.5 (91.4–108.4) | 97.3 (92.5–109.9) | 101.8 (98.7–114.6) * | 103.7 (101.8–109.8) |
| MEP_VERMA et al. [12] | 84.9 (79.7–138.5) | 84.7 (83.5–138.5) | 88.9 (85.9–97.9) | 93.7 (88.9–103.1) |

[a] Indicates mean values and standard deviation of the mean; [b] Indicates median values and percentiles 25 e 75; EG: eutrophic group; OwG: overweight group; OG: obese group; SOG: group of severely obese; Absolute frequency values (n); MIP: maximum inspiratory pressure; MEP: maximum expiratory pressure; * indicates $p < 0.05$ vs. EG in the same situation; ≠ indicates $p < 0.05$ vs. OwG in the same situation. n = 98.

However, EG and SOG presented Δ 20.9% ($p$ = 0.011), and EG e OwG presented Δ 14.8% ($p$ = 0.044) of the MIP values proposed by the equation Verma et al. [12].

In comparison, the MEP values suggested by the equation of Lanza et al. [11], EG and OG presented Δ 5.5% ($p$ = 0.039).

Indeed, when considering the total groups, RMS was higher in boys ($p < 0.01$), and there were no differences in RMS in relation to the maturational stage ($p > 0.05$).

When comparing the MIP considering the total groups, the values obtained showed Δ −32.7% and 5.9%, compared to the values proposed by the equations of Lanza et al. [11] and Verma et al. [12], respectively (Figure 2).

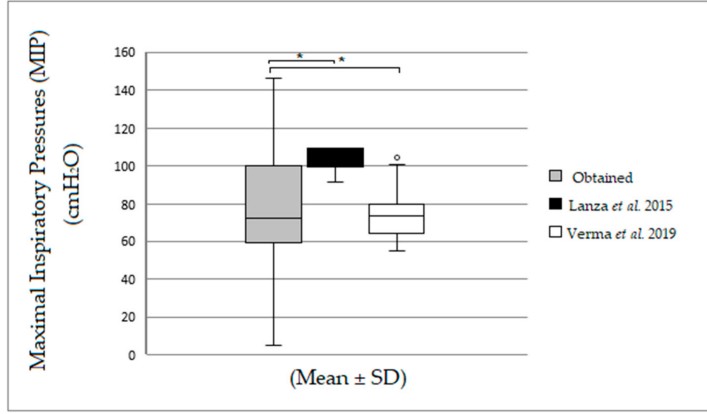

**Figure 2.** Comparison of MIP values obtained and predicted normality values [11,12]. n = 98. São Paulo-SP. * Indicating $p < 0.05$ vs. MIP obtained; ° Indicating outlier.

When comparing the MEP and considering the total groups, the values obtained showed Δ −40.7% and −42.8% compared to the values proposed by the equations of Lanza et al. [11] and Verma et al. [12]., respectively (Figure 3).

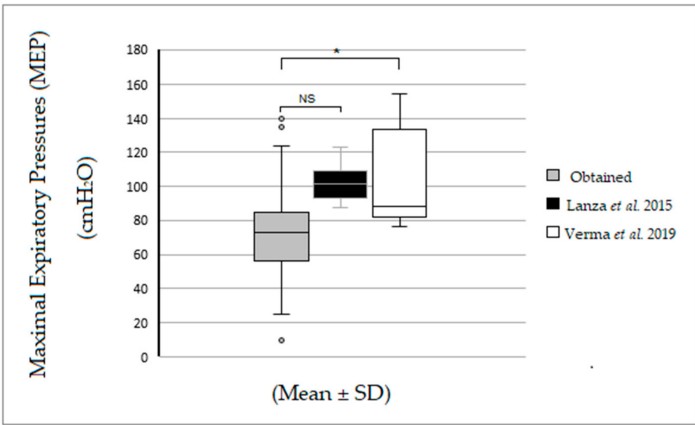

**Figure 3.** Comparison of MEP values obtained and predicted normality values [11,12]. n = 98. São Paulo-SP. * Indicating *p* < 0.05 vs. MEP obtained; NS Indicating *p* > 0.05; ° Indicating outlier.

## 4. Discussion

Since the literature has pointed to excess adiposity as one of the leading causes of decreased lung volumes and reduced RMS, the purpose of this study was to compare the measured values for RMS with those proposed by national [11] and international [12] equations described in the literature with emphasis on Brazilian adolescents with different body compositions.

The results indicated a normal RMS, regardless of nutritional and pubertal status. Thus, the results suggest that different body compositions in adolescents do not influence RMS. Furthermore, these data nullify the hypothesis initially outlined by this study that excess adiposity could reduce RMS and, consequently, lung function in adolescents. For Costa et al. [23] and Souza et al. [24], this fact could be explained by the respiratory compensation observed in overweight and/or obese individuals, which contributes to an increase in respiratory work and diaphragmatic pressure, leading to an increase in the chest during respiratory cycles [16,25].

Another factor that corroborates these findings is the association between the increase in adiposity in the respiratory muscles and consequent changes in the oxidative capacity and quality of these muscle fibers. Batista et al. [16] also added that the smaller diaphragmatic excursion caused by the accumulation of adipose tissue in the thoracic region of these individuals could also justify the results of the present study.

Regarding the higher RMS values among boys presented in this study, the literature highlights that this factor can be explained by the difference in body dimensions between boys and girls, with greater production of muscle mass in boys, as a result of higher levels of testosterone [5,7,16,17,25–27].

According to these results, there is a difference between the RMS values obtained in relation to the proposed values for both equations. It was observed in the present study that the MEP values predicted by Lanza et al. [11] and Verma et al. [12] were an overestimation. Additionally, in relation to MIP, the overestimation by Lanza et al. [11] and underestimation by Verma et al. [12] equations were also evident.

These data confirm the hypothesis in the present study showing that the mathematical equations proposed so far do not really predict RMS in adolescents with different body compositions.

For Costa et al. [23], these results can explain the fact that almost all RMS prediction equations available in the literature do not consider the use of body composition, including specific parameters such as fat mass, lean mass, and the percentage of body mass. Lanza

et al. [11] concluded that variables such as lung volume and capacity should also be considered for the prediction of RMR values. However, in their study, the lack of adequate equipment to assess lung function and body composition was highlighted as a limitation in elaborating its suggested equations.

In addition, evidence has shown that the rib cage in adolescents can adapt to a greater lack of equipment overload, as the process of rib consolidation has not been completed yet. This is a fact that allows greater thoracic mobility and lung expansion, which could lead to a positive relationship between RMS, which is related to body composition and spirometric variables. However, the lack of proper equipment in the present study to measure these parameters has also limited the exploration of these relationships [28].

Batista et al. [16] emphasized that the lack of assessment of pulmonary function, together with RMS, could mask restrictions or obstructions that may occur due to obesity, even if the respiratory pump works continuously and rhythmically, which would also explain the results of the present study.

Tenório et al. [6] and Verma et al. [12] suggest that ethnicity is also a factor to be considered in evaluating RMS, which is capable of altering lung capacity due to differences in RMS, with lung compliance and chest wall dimensions. However, this parameter was not considered in the present study. Moreover, no studies with Brazilian children and adolescents have considered variables such as body composition, lung volumes/capacities and/or ethnicity in elaborating their predictive equations yet.

Thus, further research studies regarding different ethnic groups that consider maturational aspects of different ages, nutritional status, the use of other methods of assessing body composition, spirometric measurements, as well as the standardization of collection procedures following national guidelines already established are of paramount importance for the proposition of normative reference values and equations of RMS in Brazilian children and adolescents [29].

### 4.1. Contributions to Practice

RMS evaluation is an important clinical practice for observing the health of respiratory muscles. This evaluation confirms respiratory muscle dysfunctions in different populations and pathological conditions. Thus, RMS establishes a differential diagnosis, evaluating the important response in pulmonary rehabilitation [8]. Indeed, RMS is a low-cost evaluation of easy intervention and is well-accepted in the literature. It reinforces the safety and possibility of health professionals when using this type of evaluation in clinical practice.

### 4.2. Study Limitations

Important limitations of this study should also be considered in future research: the use of digital devices to assess RMS, the assessment of lung volumes and capacities, and the measurement of anthropometric parameters (waist/hip ratio) and/or imaging tests to assess the distribution of body fat.

### 5. Conclusions

RMS was similar in adolescents with a body composition profile in different maturation stages. Adiposity did not interfere in the RMS of adolescents. Boys had higher MIP and MEP values compared to girls. The reference values proposed by the equations did not consistently match the RMS of the adolescents studied. This study reinforces the need for new studies related to RMS in order to establish normality values and a prediction equation to represent the Brazilian youth population.

**Author Contributions:** Individual author contributions are as follows: conceptualization, V.C.d.L., P.R.A. and J.M.L.A.; methodology, J.M.L.A.; investigation and intervention, V.C.d.L., P.R.A. and N.B.L.; data curation, V.C.d.L., P.R.A. and J.M.L.A.; writing—original draft preparation, V.C.d.L., P.R.A. and J.M.L.A.; writing—review and editing, V.C.d.L., N.B.L., C.S.F., T.M.d.S.S., L.R.M., M.S.F., J.M.L.A., P.S.R. and D.C.O.; supervision, M.L.M., A.H.L.J., J.M.L.A. and N.B. All authors have read and agreed to the published version of the manuscript.

**Funding:** This research received no external funding.

**Institutional Review Board Statement:** The human study protocol was approved by the Ethics Committee of Irmandade da Santa Casa de Misericórdia de São Paulo (protocol code 34634414.5.0000.5479 and date of 17 April 2017).

**Informed Consent Statement:** Informed consent was obtained from all subjects involved in the study. Written informed consent was obtained from the patient(s) to publish this paper.

**Data Availability Statement:** The data presented in this study are available on request from the corresponding author. The data are not publicly available due to privacy.

**Acknowledgments:** We thank the Pediatrics Outpatient Clinic of the Irmandade da Santa Casa de Miser-icórdia de São Paulo (ISCMSP) and the Municipal Elementary Schools (EMEF) Dona Angelina Maffei Vita, Romão Gomes, Plínio de Queiroz and Guilherme de Almeida for allowing the institution to research and evaluate their students, as well as the employees and collaborators who worked toward an accomplishment.

**Conflicts of Interest:** The authors declare no conflict of interest.

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
