# Peer review of "Respiratory Muscle Strength in Brazilian Adolescents: Impact of Body Composition"

_2673-4168, doi:10.3390/obesities3020013_

Round 1
Reviewer 1 Report
Are body fat and nutritional profile predictors of respiratory muscle power?
This study is about the comparative assessment of respiratory muscle strength (RMS) between both sexes (male and female participants). The authors want to address the consensus issue arising from use of two different equations devised by Lanza et al and Verma et al. Though the authors may have tried to their best, I have some serious reservation on the study design and the question they want to address through this work.
Major comments
1. The authors have tried to address the conflict in consensus on the use of equation that can address the conflict arising from the use of two equations proposed by Lanza and Verma et al, respectively. However, in the abstract section itself, the authors have said that “the reference values proposed by the equations do not consistently predict the RMS in the adolescents studied. This context reinforces the need for the new studies related to RMS to establish normality values and prediction equations that represent the youth public.” This indicates the fact that even this study that has proposed to address the issue remains unresolved and hence the question remains unanswered. So, I am wondering what the point of this study is. It is basically coming back to where it has started. So, could you please answer why you think this study holds merit to be published?
2. This study is not about the nutritional status as the study groups were segregated based on their BMI and fat distribution. Both nutritional status and the study design based on the BMI is completely two different things.
3. The informed consent from the participants line in the “Trial design, setting and ethics” has a line that is very vague and difficult to understand. Please rephrase that.
4. The inclusion and exclusion criteria are missing in the materials and methods section.
5. Assessment of the physical activity does not meet the international standard of explanation. Please clearly mention how the segregation of physical activity was done clinically.
Minor comments
The line number is missing throughout the manuscript and thus it becomes tricky to write the comments and/or feedbacks and to indicate their exact location.
1. Please change the topic of the manuscript from question to a topic that reflects the purpose, design and finding of the research in one line.
2. Abstract section, first line is difficult to understand, please rephrase it.
3. Do you mean “nutritional profile” in the abstract – third line? Check spelling throughout the manuscript.
4. My understanding is that the study groups were segregated based on the participants BMI and fat content of their body and not their nutritional status. Nutritional status and the BMI and/or fat content of the body is completely different thing. Please change that to reflect the appropriate study design.
5. What do you mean by predicted value in this study?
6. In the abstract section, do you mean P-value <0.05 was considered statistically significant? If so, please address that.
7. Figure 1. All flow chart for the human trial is called CONSORT diagram. Please check what are the bare minimum requirements of CONSORT diagram.
8. The English language requires extensive editing by a professional English editor.
9. The second line of “2.6 Body composition” section is very difficult to understand. Rephrase/re-write it.
10. The Body composition should be included in the “Participants” section. Merge those two sub-groups including “maturation state assessment” of the subjects. These are the parts of participants enrollment in the study and not a completely different section.
11. It is very difficult to understand the “4.2. Study limitations” as the sentences are ending abruptly and starting without capitalization.
Author Response
We appreciate the comments. We correct all of them.

Reviewer 2 Report
Comments to Author:
The research is very good and written in scientific language. It is worthy of publication in the journal
Recommendation: Minor Revision
Affiliation:
The same affiliation takes the same number- Please rearrangement.
Abstract:
L40: young
L42: nutritional profiles
L43: Do not include references in the abstract or write in full form.
L43: A cross-sectional study
L44: was carried out.
L45: (n = 44), overweight (n = 15), obese (n = 25), and severely obese (n = 14). All were submitted for anthropometric assessment.
L46: body composition analysis,
L48: Friedman was used to compare the observed vs.
L50: the RMS was higher among boys, and there were no differences in the RMS in relation to the maturational stage.
L55: nutritional statuses and
L56: Adiposity didn't interfere with adolescents' RMS.
L59: the youth population
Introduction:
L75: Santiago et al. (year)
L79: Tenório et al., (year) and all manuscript sections
Materials and Methods:
L117: H2O
L133: Figure 1
Results:
Figures 2 and 3: Clear legends
Insert X-axis and Y-axis address
Discussion:
L245: with different nutritional statuses.
Conclusions:
The RMS was similar in adolescents with different nutritional profiles and different maturation stages. Adiposity didn't interfere with adolescents RMS. Boys had higher MIP and MEP values compared to girls. The reference values proposed by the equations do not consistently predict the RMS of the adolescents studied. This context reinforces the need for new studies related to RMS to establish normality values and a prediction equation that represent the youth population.

Author Response

(The authors gave the same response as above.)

Round 2
Reviewer 1 Report
Dear authors,
Thanks for adequately addressing the comments.
I find it, now it is acceptable for publication at this stage.
Thanks and congratulations!